# Taxonomic profiling of *Nasutitermes takasagoensis* microbiota to investigate the role of termites as vectors of bacteria linked to ironwood tree decline in Guam

Garima Setia[1], Junyan Chen[1], Robert Schlub[2], Claudia Husseneder[1]*

**1** Department of Entomology, Louisiana State University Agricultural Center, Baton Rouge, LA, United States of America, **2** University of Guam, Cooperative Extension Service, Mangilao, Guam

\* chusseneder@agcenter.lsu.edu

**Data Availability Statement:** All raw FASTQ files used for the analysis are available from the NCBI database (BioProject ID PRJNA883256). URL:

## Abstract

The ironwood tree (*Casuarina equisetifolia*, family *Casuarinaceae*), an indigenous agroforestry species in Guam, has been threatened by ironwood tree decline (IWTD) since 2002. Formation of bacterial ooze by the wilt pathogen from the *Ralstonia solanacearum* species complex and wetwood bacteria (primarily *Klebsiella* species) has been linked to IWTD. In addition, termite infestation of trees was statistically associated with IWTD. Termites are known carriers of a diverse microbiome. Therefore, we hypothesized that termites could be vectors of bacteria linked to IWTD. To investigate the potential role of termites as pathogen vectors, we employed next-generation 16S rRNA gene sequencing to describe the bacteria diversity of *Nasutitermes takasagoensis* (Family Termitidae) workers collected from 42 ironwood trees of different disease stages in Guam in association with tree-, plot-, and location-related factors. *Nasutitermes takasagoensis* workers account for the majority of termite infestations of ironwood trees. The bacterial phyla composition of *N. takasagoensis* workers was typical for wood-feeding higher termites consisting mainly of Spirochaetes and Fibrobacteres. However, *Ralstonia* species were not detected and *Klebsiella* species were rare even in termites collected from trees infected with *Ralstonia* and wetwood bacteria. Feeding experiments suggested that termites prefer to consume wood with low pathogen content over wood with high pathogen load. Termites were able to ingest *Ralstonia* but *Ralstonia* could not establish itself in healthy termite bodies. We concluded that *N. takasagoensis* workers are not vectors for *Ralstonia* spp. or the bacterial endophytes associated with wetwood (*Klebsiella*, *Pantoea*, *Enterobacter*, *Citrobacter*, and *Erwinia*) that were previously observed in IWTD-infested trees. The bacterial diversity in termite samples was significantly influenced by various factors, including Tree Health, Site Management, Plot Average Decline Severity, Proportion of Dead Trees in the Plot, Proportion of Trees with Termite Damage in the Plot, Presence of Ralstonia, and Altitude.

https://www.ncbi.nlm.nih.gov/bioproject/?term=PRJNA883256

**Funding:** This research was supported by the National Institute of Food and Agriculture, U.S. Department of Agriculture, under award number 2018-38640-28418; the Western Sustainable Agriculture Research and Education (WSARE) program under project number SW19-906; the University of Guam Cooperative Extension Service; and Louisiana State University Agricultural Center. The funders had no role in study design, data collection and analysis, decision to publish, or preparation of the manuscript.

**Competing interests:** The authors have declared that no competing interests exist.

## Introduction

*Casuarina equisetifolia*, the ironwood tree, is one of the most significant agroforestry species of Guam [1, 2]. Thickets of this fast-growing tree have been a long-standing component of Guam's natural environment for thousands of years. This secondary forest species can be found across Guam primarily at landfills, beaches, and road shoulders [3, 4]. The ironwood tree possesses tolerance against salt, partial water logging, and attack by many pests and diseases [5]. This low-maintenance tree also has the ability to fix atmospheric nitrogen and control soil erosion [5]. Ironwood trees act as windbreak and shelterbelt, and thus, play an important role in the typhoon-prone island of Guam [2, 5]. Sudden decline in the health of the ironwood trees was first reported on a local farm in 2002 [1, 6]. Sick trees exhibited symptoms of yellowing and thinning of foliage, and branch die-back, along with droplets of ooze and areas of wetwood when cross-sectioned. These symptoms would become severe over time and gradually lead to tree death within 5 years. This condition has been called Ironwood Tree Decline (IWTD) [1, 7]. By 2005 IWTD had spread across the island impacting the health of thousands of Guam's ironwood trees [2]. The decline has continued to spread gradually over the past two decades [10].

Various bacteria have been isolated from ironwood trees in decline and are considered as potential pathogens associated with IWTD [8]. The bacterial vascular wilt bacterium *Ralstonia solanacearum* species complex is one of the major predictors of IWTD [9]. *Ralstonia* is known to cause wilting of the tree by colonizing the water-transporting xylem tissues and blocking water transport [10]. *Ralstonia* has been detected in the ooze and tissues from ironwood trees under decline using immunodiagnostics and has shown pathogenicity by causing wilt symptoms in healthy ironwood seedlings upon inoculation [11–13]. It has now been determined that two genospecies are associated with IWTD, *Ralstonia pseudosolanacearum* and *R. solanacearum* with the former species accounting for nearly all of the isolates [13]. Two bacterial species from the genus *Klebsiella* (*Klebsiella oxytoca* and *Klebsiella variicola*) have also been isolated from ironwood trees in decline and have been considered opportunistic pathogens for IWTD [9, 12]. *Klebsiella* species are known to cause wetwood symptoms such as brown discoloration of the central core by infecting the xylem of the tree [14, 15]. Although presence of *Klebsiella* spp. alone has not been described as a significant predictor of decline, manifestations of bacterial wetwood colonization in the form of percent wetwood and the initiation of ooze were significantly associated with IWTD [12]. Bacteria from genera *Pantoea*, *Enterobacter*, *Citrobacter*, *Erwinia*, and *Kosakonia* were also isolated from the ooze of ironwood trees in decline but occurred less frequently than *Ralstonia* and *Klebsiella* spp. [9].

Insects are considered the most common vectors for the transmission of plant pathogens [16–18]. Beetles (*Protaetia pryeri* (Janson) and *Protaetia orientalis* (Gory and Percheron)), gall wasps (genus *Selitrichodes*), as well as termites, have been found attacking ironwood trees in Guam [2]. Beetles and gall wasps were eliminated from the list of potential factors responsible for IWTD [2, 19]. The presence of termites, however, was found to be significantly associated with IWTD [9, 20]. The termites attacking ironwood trees in Guam were identified as a species from the *Nasutitermes takasagoensis* (Nawa) (Blattodea: Termitidae) complex, *Coptotermes gestroi* (Wasmann) (Blattodea: Rhinotermitidae), *Microcerotermes crassus* Snyder (Blattodea: Termitidae) and an unknown *Microcerotermes* species by morphology and the closest match of their DNA barcodes to reference sequences in NCBI GenBank [21]. Over 90% of ironwood trees in Guam that had an infestation with live termites were attacked by *N. takasagoensis* foragers [21].

The termite species *N. takasagoensis* belongs to the family Termitidae, which comprises the so-called "higher termites" [22, 23]. Wood-feeding higher termites form associations with

bacteria for the degradation of lignocellulose that they consume and for other functions such as nitrogen fixation [23, 24]. Spirochaetes, Fibrobacteres, Bacteroidetes, Firmicutes, Proteobacteria, Actinobacteria, and Margulisbacteria (TG3 phylum) are among the core bacterial phyla present in *N. takasagoensis* [25–30]. Some of these bacteria are obligate symbionts that co-evolved with their termite host to form a co-dependent mutualistic symbiotic association [23, 31]. Other bacteria present in termites are facultative symbionts acquired from the environment, e.g., with consumed plant matter or through soil contact.

It is conceivable that termites attacking trees suffering from IWTD come in contact with or ingest wood infested with putative IWTD pathogens. While foraging, termites can possibly transmit these putative IWTD pathogens to trees that are not in decline. Thus, we hypothesized that members of *N. takasagoensis*, which represents the major termite species attacking ironwood trees, are vectors for pathogenic bacteria associated with IWTD. The objectives of this study were to (a) describe the bacterial taxa associated with *N. takasagoensis* workers collected from healthy and sick ironwood trees in Guam to test if these workers carry putative pathogens associated with IWTD, (b) evaluate the relationship between the tree-, plot-, and location-related factors associated with ironwood trees attacked by *N. takasagoensis* workers and bacterial diversity of those *N. takasagoensis* worker samples, (c) determine if *N. takasagoensis* workers prefer food sources with low over those with high pathogen content, and (d) determine the ability of *R. solanacearum* bacteria to survive in the body of termite workers.

## Methods

### Samples and metadata

**Termite samples.** Forty-two *N. takasagoensis* termite samples were collected from ironwood trees in 2019–20 (S1 Fig). These samples, which included both soldier and worker termites, were partitioned and stored in 70% ethanol for morphological identification and 95% ethanol for Illumina sequencing. The morphological species identification was performed by diagnosing the characteristic features of soldiers based on published keys [32, 33] using a stereo microscope (Leica MZ16). Metadata for tree-related, plot-related, and location-related factors were recorded for each termite sample (S1 Table).

**Tree-related factors.** Each tree from which termite samples were collected was assayed for presence of *Ralstonia* using *R. solanacearum*-specific immunodiagnostic test kit (Agdia, Inc. Indiana. U.S.A.) [12]. This kit screens plant samples for the presence of *R. solanacearum* using an antigen-antibody-based test but does not measure concentration. Decline Severity (DS) of ironwood trees in Guam was measured by visually assigning trees to five categories based on the level of damage and fullness of branches at the time termites were sampled ranging from DS = 0 for trees with no symptoms (symptomless), DS = 1 with few symptoms (slightly damaged), DS = 2 with clearly visible disease symptoms (distinctly damaged), DS = 3 with severe disease symptoms (heavily damaged), to DS = 4 for trees that had shed almost all their foliage and were about to die (nearly dead) [2, 20]. The assessment of Tree Health was further simplified by categorizing ironwood trees as healthy if they showed no symptoms (DS = 0) or sick if they showed symptoms (DS = 1 through 4).

**Plot-related factors.** Plot Average DS is a measurement of the extent to which the health of the trees in a plot is deteriorating. A plot refers to a circular region with a tagged termite sample tree located at its center and a radius of approximately 30 meters. To calculate the Plot Average DS, the DS values of all live trees within the plot were added together and the sum was divided by the number of live trees within that plot. Proportion of Dead Trees in Plot was determined by dividing the number of dead trees by the total number of trees within a 30 meters plot radius. Proportion of Trees with Termites in Plot provides the proportion of live

trees that are affected by termites within that specific area. It was determined by dividing the number of live trees with termite activity by the total number of live trees within a given plot. Termite activity was detected by the presence of termite nests or tunnels on the tree.

**Location-related factors.** Geographical Location is the specific area in Guam where the ironwood tree was located from which a termite sample was taken. Altitude of the tree location was classified relative to the mean sea level measured in meters at the base of the tree ("low" $\leq$ 100m, "high" >100m). Parent Material refers to the unconsolidated, relatively unweathered minerals or organic matter (source material) that has contributed to the formation of soil at the location where a tree is growing. The information regarding Parent Material was obtained from the Natural Resources Conservation Service of the United States Department of Agriculture (https://websoilsurvey.sc.egov.usda.gov/App/WebSoilSurvey.aspx). The Parent Material was categorized into three types: coralline limestone (referred to as "lime"), residuum derived from tuff and tuff breccia (referred to as "tuff"), and water-deposited coral sand (referred to as "sand"). Site Management refers to the level of human maintenance that is applied to a specific area. Unmanaged sites, such as abandoned lots, were classified as not managed, sites that are moderately managed, such as farm windrows, were classified as moderately managed, and well-managed sites, such as golf courses, were classified as highly managed.

## DNA extraction, primer selection and Illumina sequencing

Five termite workers were pooled from each sample to extract DNA using the DNeasy Blood & Tissue kit (Qiagen, Germantown, MA). The DNA concentration was measured using an Invitrogen Qubit 4 Fluorometer (Thermo Fisher Scientific, Wilmington, DE) and the Qubit dsDNA BR Assay Kit (Invitrogen™, Life Technologies™). Finally, 20 μl aliquots with 2.5 ng/μl DNA concentration per sample were sent to the University of New Hampshire Hubbard Center for Genome Studies for library preparation and Illumina sequencing. The V1-V3 hyper variable region of the bacterial 16S rRNA gene was amplified from the DNA samples using a forward (27F) and two reverse primers (519Rmod and 519Rmodbio) to capture a broad range of biodiversity [34, 35]. The PCR products were then sequenced on the 2x250bp Illumina NovaSeq platform using the Illumina Nextera Dilute library protocol (Illumina, San Diego, CA).

## Bioinformatics and statistical analysis

The sequence data analysis was performed using the Quantitative Insights into Microbial Ecology (QIIME2) pipeline version 2021–4 [36, 37]. The demultiplexed sequencing reads (FASTQ format) were subjected to quality control using the DADA2 plugin in QIIME2 for denoising and chimera removal [38]. All the sequencing reads were of high quality (Phred quality score >30). Further analysis was performed using only the forward reads as there was a substantial number of forward and reverse reads that did not overlap adequately. The output of the DADA2 procedure was a table containing Amplicon Sequence Variants (ASVs) that represent unique sequences and a representative sequence file showing all the sequences of the ASVs. The sequences produced from the same sample using two different reverse primer sets were merged. The raw sequence data used in this study were submitted to NCBI GenBank (BioProject ID PRJNA883256).

Sequence depth-based alpha rarefaction curves were generated using QIIME2 by plotting the number of ASVs (ASV richness), Faith's phylogenetic distance (Faith's PD) between the ASVs, and richness of ASVs based on their evenness (Shannon diversity) against the sequencing depth. Alpha rarefaction curves based on sample size and coverage were plotted using the R package iNEXT (iNterpolation/ EXTrapolation) [39]. For sample size and coverage-based rarefaction curves, Hill numbers (q) including ASV richness ($q = 0$), Shannon diversity ($q = 1$),

and Simpson diversity ($q$ = 2) were used to determine the effective diversity of ASVs [39, 40]. The iNEXT package was also used to extrapolate effective diversity when the sample size is doubled.

The ASVs were taxonomically assigned through SILVA 132 reference database for 16S rRNA genes [41] using the BLAST algorithm [42] with a 97% pairwise identity cutoff. The ASVs with less than 97% sequence identity to the reference sequences in the SILVA database were deemed unassigned and filtered from the ASV table. The relative abundance of taxa within each sample was visualized using bar plots generated from the filtered ASV table. The 20 ASVs with the highest number of reads from all the samples were further assigned to the top BLAST hit in the NCBI GenBank database using BLAST+ algorithm for confirmation.

Diversity analyses were computed in QIIME2 after subsampling all samples to a common depth of 15,951 sequences using only taxonomically assigned ASVs for the rarefaction procedure. Alpha diversity indices (ASV richness, Faith's PD, Pielou's evenness, and Shannon diversity) of the microbial community were computed for each sample. The impact of the various tree-, plot- and location- related factors on these four bacterial alpha-diversity metrics was tested with two methods. Significant effects of factors with categorical data (Presence of *Ralstonia*, Tree DS, Tree Health, Location, Altitude, Parent Material, and Site Management) on bacterial diversity were determined with Kruskal-Wallis ANOVA (H) [43], followed by false discovery rate correction using Benjamini-Hochberg procedure [44]. Correlation between bacterial diversity and numerical data (Plot Average DS, Proportion of Dead Trees in Plot, and Proportion of Trees with Termites in Plot) was determined using Spearman rank ($r_s$) tests.

Beta diversity was computed using weighted Unifrac distance metric [45]. Beta-group significance was calculated using Permutational Multivariate Analysis of Variance (PERMANOVA, 999 permutations) [46, 47]. The homogeneity of variance of the bacteria composition among groups of termite samples determined by multivariate spread was assessed using the PERMDISP test at 1000 permutations [48]. Factors shown to be significant using one-factorial PERMANOVA tests were further tested using a multifactorial PERMANOVA test, also known as ADONIS [46] to confirm their significance and determine the interactions between these factors. The differences in the abundance of taxa that caused the significant differences among groups within beta diversity factors were determined using the Analysis of Compositions of Microbiomes with Bias Correction (ANCOM-BC) in QIIME2 [49].

### Feeding experiments to assess consumption of *R. solanacearum* by *N. takasagoensis* workers

**Termite collection for feeding experiments.** Three experiments were conducted to test if *N. takasagoensis* workers would show any feeding preference in relation to pathogen load of the food source and if *Ralstonia* would be able to survive in termites. Termites were subjected to (1) four-choice tests between wood pieces with different putative pathogen loads from sick and healthy ironwood trees, (2) two-choice tests between wood from healthy ironwood tree inoculated with *Ralstonia* and control with no *Ralstonia* inoculation, and (3) no-choice tests with *Ralstonia* inoculated filter paper for examining *Ralstonia* bacteria's ability to survive in *N. takasagoensis* workers. For these tests, the *R. solanacearum* species complex isolate 19–147 was used [13]. This *R. pseudosolanacearum* isolate was a subculture of the original isolate obtained from the bacterial ooze from a root section of a heavily damaged (DS = 3) ironwood tree from Guam and successfully purified at the University of Hawaii in 2019 [13]. Sections from three *N. takasagoensis* nests located at Bernard Watson's farm (GPS coordinates: 13˚56702', 144˚.87746'), Mangilao Golf Couse (13˚47111', 144˚8452') and UOG Yigo Station (13˚53308', 144˚87222') were transported to the lab and immediately dissected to extract the termites for

conducting four-choice tests and two-choice tests. Logs of ironwood trees containing *N. takasagoensis* workers and soldiers were brought to the lab from UOG Campus (13˚43020', 144˚ 80008'), UOG Yigo Station (13˚53356', 144˚87116') and UOG Yigo Station (13˚53288', 144˚ 87163') for conducting no-choice tests. The experimental units were set up on the day of collection and maintained in the dark at 26 ± 2˚C. Dead termites were counted daily and removed from the experimental units.

**Four-choice tests among *C. equisetifolia* wood pieces with different amounts of *R. solanacearum* and wetwood bacteria.** To obtain the wood pieces for the four-choice test, four *C. equisetifolia* trees were cut, each with either positive or negative test results for *R. solanacearum* and low or high amounts of wetwood bacteria. The tree trunks were cut and sliced into roughly 2 cm thick disks using a circular saw. To maximize the recovery of wetwood tissue, wood pieces of approximately 0.5x2x1.5 cm$^3$ were cut from the center of the disks using a chisel and hammer. Trees with no wetwood stains were considered to have low levels of wetwood bacteria and trees with visible dark stains were considered to have high levels of wetwood bacteria. Presence of *R. solanacearum* bacteria was determined using immunodiagnostic test kits (Agdia, Inc., Indiana, U.S.A.). The four treatments used for four-choice test were: (1) *R. solanacearum* negative and low wetwood bacteria (Tree location: Yigo Experiment Station (13˚ 533523', 144˚871081')), (2) *R. solanacearum* positive and low wetwood bacteria (Yigo Experiment Station (13˚533523', 144˚871130')), (3) *R. solanacearum* negative and high wetwood bacteria (Bernard Watson's Farm (13˚34.026', 144˚52.584')), and (4) *R. solanacearum* positive and high wetwood bacteria (Bernard Watson's Farm (13˚34.028', 144˚52.597')).

The initial weight of each wood piece was recorded. The four wood pieces were equally spaced in a 145x20mm Petri dish filled with sand at a 12% moisture level. The experiment was organized in a randomized complete block design with a total of 15 blocks and five replicates from each of the three termite colonies. A total of 300 workers and 60 soldiers of *N. takasagoensis* were placed in each Petri dish. The final weight of the four pieces of wood after a three-week period was subtracted from the initial weight to measure consumption. Change in average weight of the wood pieces was tested for significance among the treatments through a one-way analysis of variance followed by a Tukey's Studentized Range test for post-hoc analysis, at a significance level of α< 0.05 using SAS Software (SAS 9.4).

**Two-choice test comparing consumption of wood soaked with *Ralstonia* overnight culture to a saline control.** The two-choice tests aimed to determine the consumption of *N. takasagoensis* workers for *C. equisetifolia* wood inoculated with *R. solanacearum* versus saline-inoculated wood. An overnight sub-culture of *Ralstonia* isolate 19–147, which was confirmed through pathogenicity tests and sequencing [50], was created in a shaker-incubator set at 220 rpm and 28˚C using Casamino Acid-Peptone-Glucose (CPG) broth medium (1 g Casamino acid (casein hydrolysate), 10 g Peptone, and 5 g Glucose in 1 liter distilled water) [51]. Serial dilutions ranging from $10^{-1}$ to $10^{-10}$ were then made using 0.85% saline. The optical density of the overnight culture was measured to be 2.5 ($OD_{600}$) using the SPECTRONIC™ 200 spectrophotometer (Thermo Scientific). The number of colony-forming units in the culture was found to be 9.251E+9 CFU/ml through plating on CPG agar plates (1g casamino acid, 10g peptone, 5g glucose, and 17g agar in 1 liter distilled water autoclaved at 121˚C for 20 minutes). We used dilutions of $10^{-4}$, $10^{-6}$, and $10^{-8}$ for the two-choice tests because based on the results of a pilot study performed on *Coptotermes formosanus* workers using the *R. solanacearum* strain GMI1000 from the American Type Culture Collection (ATCC), the presence of bacteria did not repel or negatively impact the survival of workers at these dilutions [50].

A healthy ironwood tree (*R. solanacearum* negative and no evidence of wetwood staining) was cut with a circular saw to obtain wood pieces (approximately 0.5x2x1.5 cm$^3$) for two-choice tests. These wood pieces were dried in an oven for 2 days at 100˚C. The initial weight of

the dried wood pieces was recorded. Wood pieces were inoculated with 200 μL of *R. solanacearum* bacterial culture at three different dilutions ($10^{-4}$, $10^{-6}$ and $10^{-8}$) and 0.85% saline as control treatment. The inoculated wood pieces were kept in Petri dishes (60x20mm) filled with sand (12% moisture) and placed in an experimental set up following a randomized complete block design. The study consisted of 45 experimental units combining three colonies, three concentrations, and five replicates. One hundred workers and 20 soldiers of *N. takasagoensis* were introduced into each replicate. After three weeks, the weight of each wood piece was recorded after drying. The results were then analyzed as described for the four-choice test.

**No-choice tests to measure ingestion and survival of *Ralstonia* in termite guts.** The experiment was conducted using a randomized complete block design with three colonies, each consisting of five replicates in separate Petri dishes along with controls for each of them. Filter paper was soaked with 100 μL of *Ralstonia* (isolate 19–147) culture diluted in 0.85% saline at concentrations of $10^{-4}$, $10^{-6}$, or $10^{-8}$, and 0.85% saline without any *Ralstonia* as negative control. A total of 50 workers and 5 soldiers per *N. takasagoensis* colony were placed into 60x20 mm Petri dishes and were made to feed on the *R. solanacearum* bacterial dilutions ($10^{-4}$, $10^{-6}$, $10^{-8}$) or the control for three time periods (2 days, 4 days, and 6 days), with different sets of Petri dishes being used for each time period. This initial six-day phase of the no-choice test was referred to as Phase 1. The termites that had fed for the longest time (6 days) on the *Ralstonia* concentrations in Phase 1 were then transferred to filter paper without *Ralstonia* for an additional 2 days (Phase 2), to determine if ingested *Ralstonia* would survive. After 2, 4 and 6 days into Phase 1, as well as after 8 days (6 days of Phase 1 and 2 days of Phase 2), 8 workers were removed from each Petri dish and stored in 95% ethanol for subsequent sequencing to measure the presence and change in abundance of *Ralstonia* in the termites over time. The DNA was extracted from a pool of five workers from each sample (DNeasy Blood & Tissue kit, Qiagen, Germantown, MA). The V4 region of the 16S rRNA gene of the bacterial DNA was amplified using 515F and 926R primers [36, 52], and sequenced on the Illumina NovaSeq platform (2x250bp) at the University of New Hampshire Hubbard Center for Genome Studies using the Illumina Nextera Dilute library protocol (Illumina, San Diego, CA). The QIIME2 pipeline was used for taxonomic assignment and diversity analysis as described above.

## Results

### Number of sequence reads and ASVs

A total of 11,106,360 raw sequences were obtained from the 42 *N. takasagoensis* samples. After DADA2 quality filtering and removal of chimeras, 9,902,718 sequence reads and 12,903 ASVs were obtained. The ASVs that did not have a reference in the SILVA database with at least 97% similarity were removed, resulting in 1,709,419 reads and 462 ASVs across all samples. Only 3.5% of the total ASVs had a taxonomic assignment at a 97% identity cutoff. The minimum sequencing depth common to all samples was reduced from 50,410 to 15,951 after removing the unassigned ASVs.

### Sequence depth-, sample- and coverage-based rarefaction

The rarefaction curves for sequence-depth based on ASV richness and Faith's PD indices began to plateau after reaching a sequencing depth of 10,000 to 15,000. The Shannon diversity rarefaction curve levelled out at a depth of less than 5,000 sequences (Fig 1A). This suggests that the sequencing depth used in our study was adequate to capture the majority of taxa and acquiring more sequences would not result in a considerable increase in diversity.

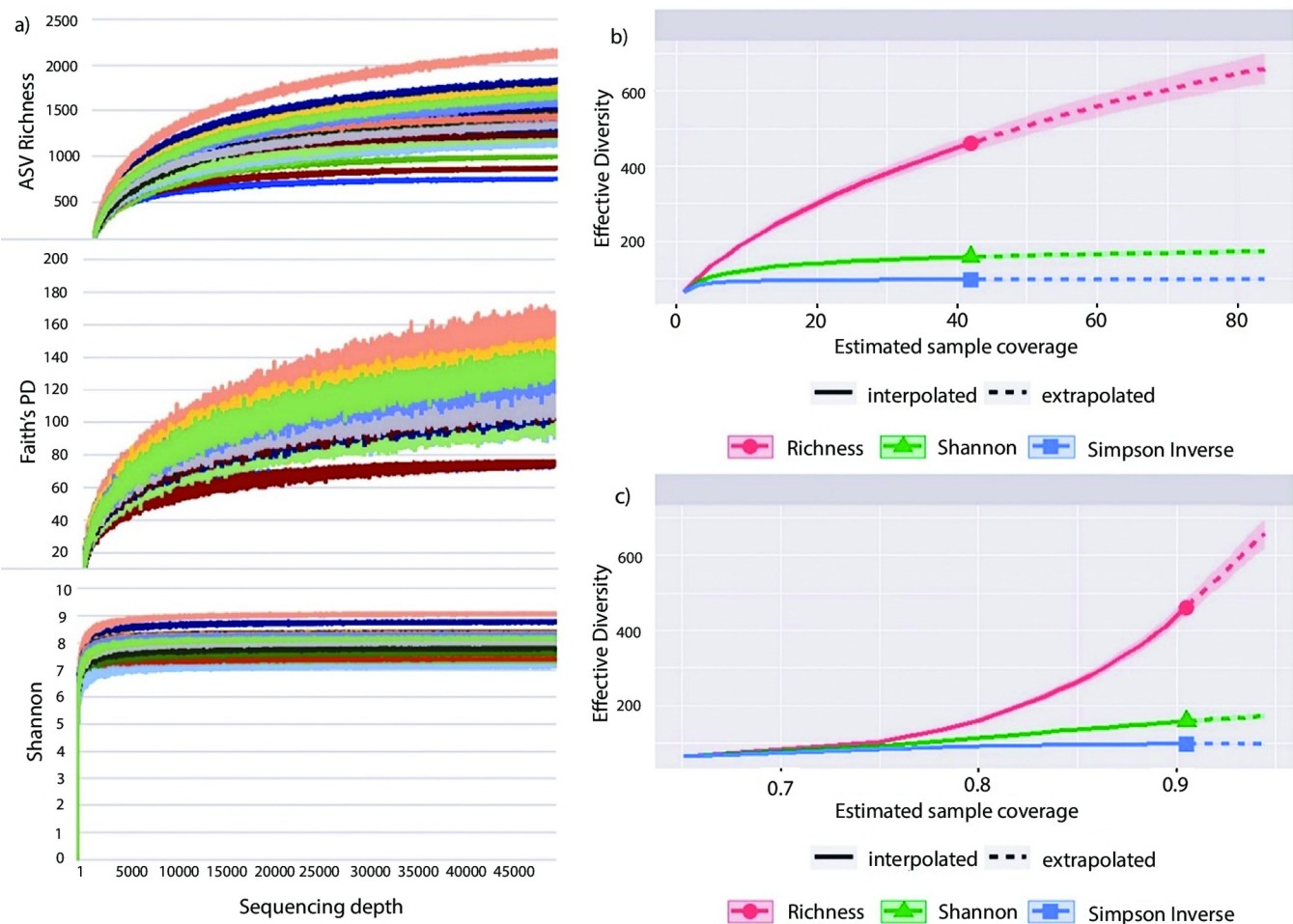

**Fig 1. Bacterial diversity rarefaction in *Nasutitermes takasagoensis* worker samples.** a) Sequence-based rarefaction curves of bacteria diversity showing the number of ASVs, Faith's phylogenetic distance and Shannon diversity indices in 42 samples of *Nasutitermes takasagoensis* workers plotted against sequencing depth. b) Sample-based rarefaction curves with effective bacterial diversity for different metrics plotted against the number of samples. c) Coverage-based rarefaction curves with effective diversity plotted against estimated sample coverage. Solid lines indicate intrapolation up to the actual sample size; dashed lines represent extrapolation to twice the sample size.

The sample-based rarefaction (Fig 1B) demonstrated that the curves for the Shannon and Simpson inverse indices plateaued at an effective diversity of 150 and 100, respectively, and extrapolating the curve to double the sample size did not result in an increase in the diversity captured. For ASV richness, the curve began to level off but did not reach an asymptote at 42 samples. If the sample size were doubled, the number of ASVs would increase from 462 to over 600 (Fig 1B). Nevertheless, the additional richness would mainly comprise rare ASVs, as the Shannon diversity and Simpson inverse index did not demonstrate an increase with the inclusion of more ASVs.

The coverage-based rarefaction (Fig 1C) illustrates effective diversity with respect to sample completeness. The interpolated portions of the rarefaction curves based on coverage reached a sample coverage of more than 90% at an ASV richness of 462, a Shannon diversity of about 150, and a Simpson inverse of approximately 100 (Fig 1C). Extending these curves to nearly 95% sample coverage through extrapolation led to an increase in richness to above 600. However, the Shannon diversity and Simpson inverse only experienced an incremental increase (Fig 1C).

## Taxa composition

Twenty-two bacterial phyla were identified in the 42 *N. takasagoensis* worker samples collected from sick and healthy ironwood trees in Guam (Fig 2 and S1 Table). Spirochaetes (48.16%), Fibrobacteres (41.38%), Bacteroidetes (3.61%), Proteobacteria (3.38%), Margulisbacteria (0.84%), Acidobacteria (0.77%), Planctomycetes (0.65%), Actinobacteria (0.47%), Synergistetes (0.19%), Firmicutes (0.19%), Tenericutes (0.18%) and Chloroflexi (0.06%) were present in all 42 samples with the major phyla of Spirochaetes and Fibrobacteres combined accounting for almost 90% of the bacterial community. The remaining 10 phyla combined accounted for ≤0.04% and were present in 26 samples and less (S2 Table).

Among the 462 ASVs analyzed, the top 20 with the highest number of reads were found in all 42 samples (S3 and S4 Tables). The most commonly detected ASVs (>1,000 reads), found in all the samples, were assigned to the dominant phyla Spirochaetes (two ASVs from the genus Treponema and one uncultured bacterium) and Fibrobacteres (two uncultured Fibrobacteres and one Chitinivibrionia) (S4 Table). The most dominant genus present in the *N.*

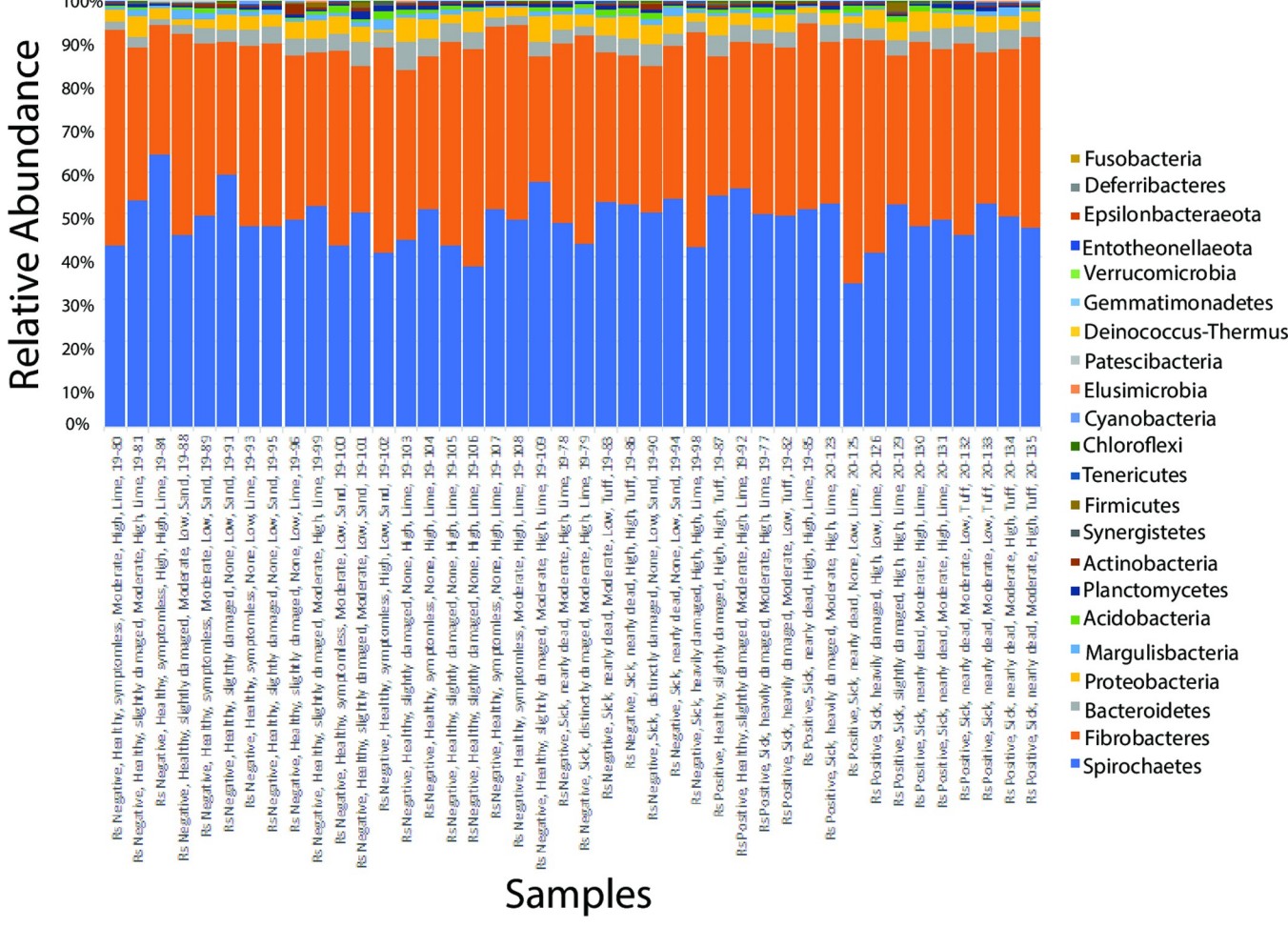

**Fig 2. Relative abundance of bacterial phyla associated with 42 samples of *N. takasagoensis* workers collected from ironwood trees in Guam.** Phyla are shown in decreasing abundance from bottom to top. The sample name on the x-axis encodes the following seven factors: (1) presence (Positive) or absence (Negative) of *Ralstonia* (Rs), (2) Tree Health (Healthy or Sick), (3) Tree DS (symptomless (DS = 0), slightly damaged (DS = 1), distinctly damaged (DS = 2), heavily damaged (DS = 3), and nearly dead (DS = 4)), (5) level of Site Management (None, Moderate, High), (6) Altitude (High, Low) and (7) Parent Material (Sand, Lime, Tuff).

*takasagoensis* workers was Treponema of the phylum Spirochaetes. In addition, an uncultured delta proteobacterium, which belonged to the phylum Proteobacteria, was also found to average over 1,000 reads per sample. The top 20 ASVs present in most *N. takasagoensis* samples but with reads averaging less than 1,000 per samples were an uncultured candidate division ZB3 bacterium (phylum Margulisbacteria), two uncultured Bacteroidetes bacteria (phylum Bacteroidetes), an uncultured Chitinivibrionia bacterium (phylum Fibrobacteres), uncultured Acidobacteria (phylum Acidobacteria), uncultured planctomycete (phylum Planctomycete), an uncultured Alphaproteobacteria bacterium (phylum Proteobacteria), three uncultured Bacteroidetes bacterium (phylum Bacteroidetes) and an uncultured Spirochaetes bacterium (phylum Spirochaetes) (S4 Table).

One of the objectives of this study was to investigate if bacteria linked to IWTD were present in termites collected from ironwood trees in Guam. The taxonomic analysis found no evidence of *Ralstonia* spp in *N. takasagoensis* workers, regardless of the health of the ironwood tree the termites were collected from. However, various genera, named *Comamonas*, *Duganella*, *Hydrogenophaga*, *Lautropia*, *Massilia*, *Ottowia*, *Parapusillimonas* and *Xenophilus* from the same family (Burkholderiaceae) as the genus *Ralstonia* were present in the termite samples. Genus *Klebsiella* (unidentified species) was detected in only four samples in minor abundance ($\leq$ 5 sequence reads). No other putative pathogens, such as *Pantoea*, *Enterobacter*, *Citrobacter*, *Erwinia*, and *Kosakonia* were detected [9]. The absence of *Ralstonia* and the limited occurrence and quantity of *Klebsiella* implies that *N. takasagoensis* workers are not likely to act as a carrier for these putative pathogens linked with IWTD.

## Alpha diversity

Presence of *Ralstonia*, Tree DS, Location, Altitude, and Parent Material of the tree did not have any significant effects on the alpha diversity indices of bacterial communities of termites. Although there were no significant differences in alpha diversity across the five DS stages, significant differences were observed when Tree DS categories were reduced to sick (DS 1–4) and healthy (DS 0) trees. The bacterial communities in termites collected from sick trees had significantly greater phylogenetic distances than those collected from healthy trees, as demonstrated by Faith's PD (p = 0.02, H = 5.07, Kruskal-Wallis ANOVA, Fig 3A). However, there were no significant differences in the ASV Richness, Pielou's evenness, and Shannon diversity between the bacterial communities of termites collected from healthy and sick trees.

The level of Site Management had a significant effect on the evenness and Shannon diversity of the bacterial community associated with termites. Specifically, intense Site Management resulted in a lower Pielou's evenness and lower Shannon diversity compared to moderate and high management levels (Pielou's evenness, p = 0.04, H = 6.17; Shannon diversity, p = 0.02, H = 7.41, Kruskal-Wallis ANOVA, Fig 3B and 3C, respectively). However, Site Management did not show a significant impact on the ASV Richness and Faith's PD of the bacterial community associated with termites.

An increase in plot decline was found to be associated negatively with some aspects of the bacterial diversity of the termites in certain plot-related factors. Evenness of the bacteria community within termite samples was negatively correlated with Plot Average DS (Pielou's evenness, p = 0.01, $r_s$ = -0.31, Spearman's Rank test, n = 42, Fig 3D). A negative correlation was also observed between Shannon diversity of bacterial communities within termite samples and higher Proportion of Dead Trees in Plots (Shannon diversity, p = 0.03, $r_s$ = -0.32, Spearman's Rank test, n = 42, Fig 3E). Proportion of Dead Trees in Plots did not show significant correlation to the Pielou's evenness, ASV richness, and Faith's PD of the bacteria community in termites. With an increase in the Proportion of Trees with Termites in Plot, a significant decrease

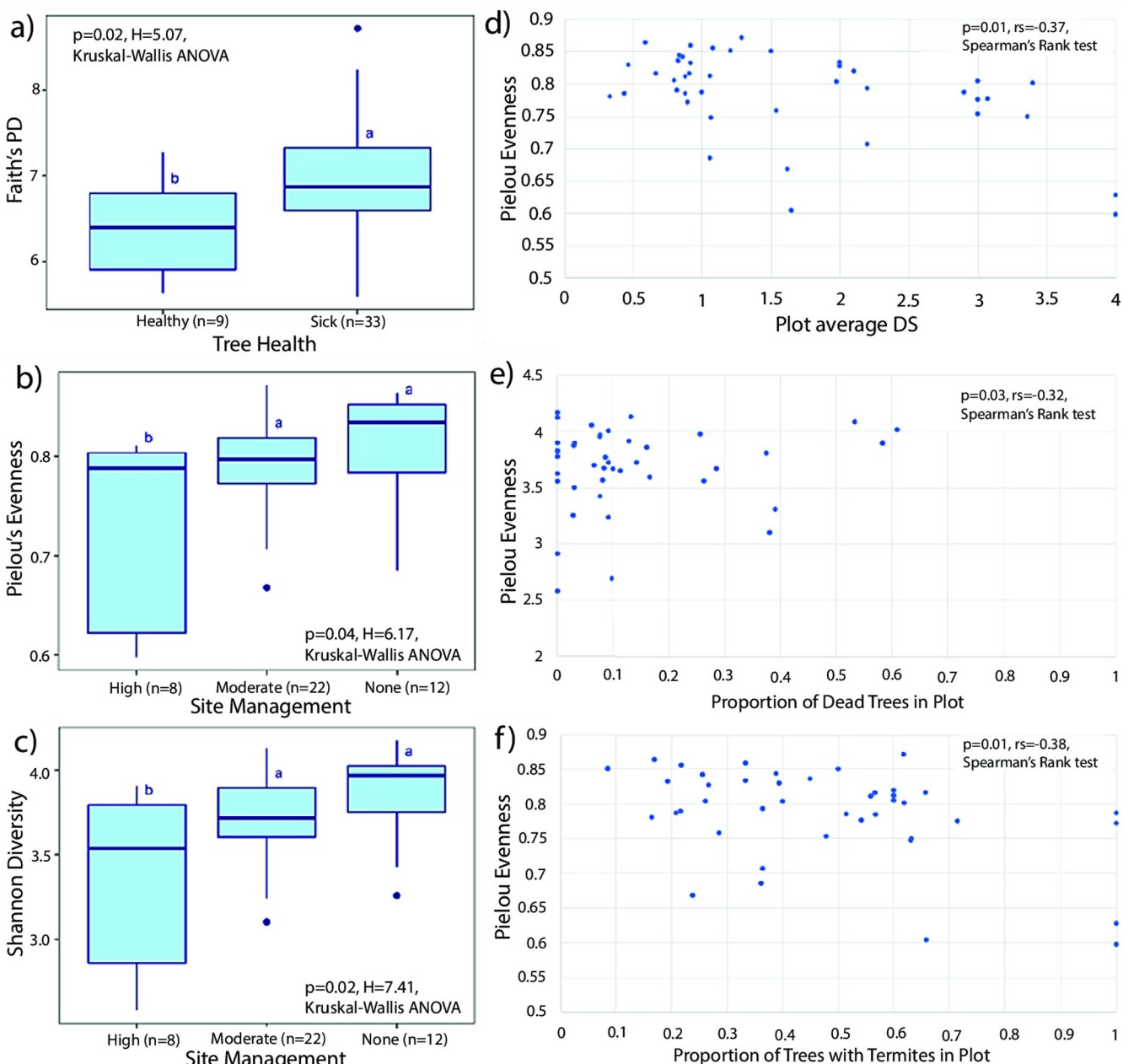

**Fig 3. Tree-, plot and location-related factors with significant effects on different aspects of alpha diversity of termite bacteria communities.** Different letters indicate significant difference. a) Faith's PD of bacteria communities of termites collected from healthy and sick ironwood trees. b) Pielou's evenness and c) Shannon diversity index of bacterial communities of termites collected from highly, moderately or non-managed sites. d) Correlation between Pielou's evenness of termite bacterial communities and Plot Average DS. e) Correlation between Shannon diversity of bacterial communities of termites and Proportion of Dead Trees in Plots. f) Correlation between Faith's PD of bacterial communities of termites and Proportion of Trees with Termites in Plot.

in the evenness of the bacterial community of termites feeding on those trees was observed (Pielou's evenness, $p = 0.01$, $r_s = -0.38$, $n = 42$, Spearman's Rank test, Fig 3F), but ASV richness, Shannon diversity, and Faith's PD were not significantly impacted by Proportion of Dead Trees in Plot.

## Beta diversity

Presence of *Ralstonia*, Altitude and Parent Material had a significant impact on the diversity of bacterial communities in termites, as shown by the single factor PERMANOVA (p≤0.03, S5 Table). The bacterial community of termite samples taken from trees growing on Lime and Sand, as well as Sand and Tuff, were significantly different (both p = 0.01), but there was no significant difference between Lime and Tuff samples. The PERMDISP analysis confirmed that the variance in bacterial communities was homogeneous across the three factors (S5 Table), allowing for a multifactorial ADONIS test to be performed.

Presence of *Ralstonia*, and Altitude showed significant effects (Pr(>F)≤0.04) on bacteria diversity in termites based on the ADONIS test results (Table 1). While the single-factor PER-MANOVA analysis showed that Parent Material had a significant impact on the diversity of bacterial communities in termites, the multifactorial ADONIS test did not find significant effects for this factor. Instead, *Ralstonia* had the strongest effect on the variation in bacterial communities (explaining 8% of the variation in the dataset), followed by Altitude (6%). The interaction between *Ralstonia* and Parent Material showed only marginal significance, with a low R2 value, while no significant interactions were found for other pairwise combinations (Table 1).

## Differential abundance

An ASV within phylum Bacteroidetes (adjusted p value = 0.01) and an ASV within phylum Spirochaetes (Genus *Treponema*) (adjusted p value = 0.03) showed a significantly higher abundance in termites collected from *Ralstonia* negative trees as compared to *Ralstonia* positive trees using ANCOM-BC (S6A Table). The ANCOM-BC analysis also revealed that two ASVs belonging to the genus *Treponema* within the phylum Spirochaetes were significantly enriched in the bacterial communities of termites collected from trees growing at high altitudes (adjusted p values of 0.01 and 0.04, respectively) (S6B Table). Furthermore, seven ASVs were differentially abundant between the termite samples collected from sand compared to lime (all adjusted p values < 0.05), and 24 ASVs were differentially abundant between the termite samples collected from tuff compared to lime (all adjusted p values < 0.05) (S6C Table).

## Consumption of wood pieces with different levels of *Ralstonia* and wetwood bacteria by *N. takasagoensis* workers

Since we did not detect IWTD pathogens in significant amounts during the taxonomic profiling of bacterial communities in *N. takasagoensis* termite samples we wanted to investigate

**Table 1. Adonis test results showing factors affecting the differentiation of bacteria composition among groups of termite samples and their interactions.** Asterisks indicate significant effects.

| Factor | Df | Sums Of Squares | Mean Squares | F.Model | R$^2$ | Pr(>F) |
|---|---|---|---|---|---|---|
| Presence of *Ralstonia* (Positive, Negative) | 1 | 0.18 | 0.18 | 3.46 | 0.08 | 0.02* |
| Altitude (Low, High) | 1 | 0.13 | 0.13 | 2.54 | 0.06 | 0.04* |
| Parent Material (Lime, Sand, Tuff) | 2 | 0.15 | 0.08 | 1.46 | 0.06 | 0.17 |
| Presence of *Ralstonia*: Altitude | 1 | 0.02 | 0.02 | 0.47 | 0.01 | 0.75 |
| Presence of *Ralstonia*: Parent Material | 1 | 0.12 | 0.12 | 2.31 | 0.05 | 0.09 |
| Altitude: Parent Material | 1 | 0.04 | 0.04 | 0.73 | 0.02 | 0.54 |
| Presence of *Ralstonia*: Altitude: Parent Material | 1 | 0.01 | 0.01 | 0.22 | 0.01 | 0.92 |
| Residuals | 33 | 1.73 | 0.052 | | 0.72 | |
| Total | 41 | 2.39 | | | 1 | |

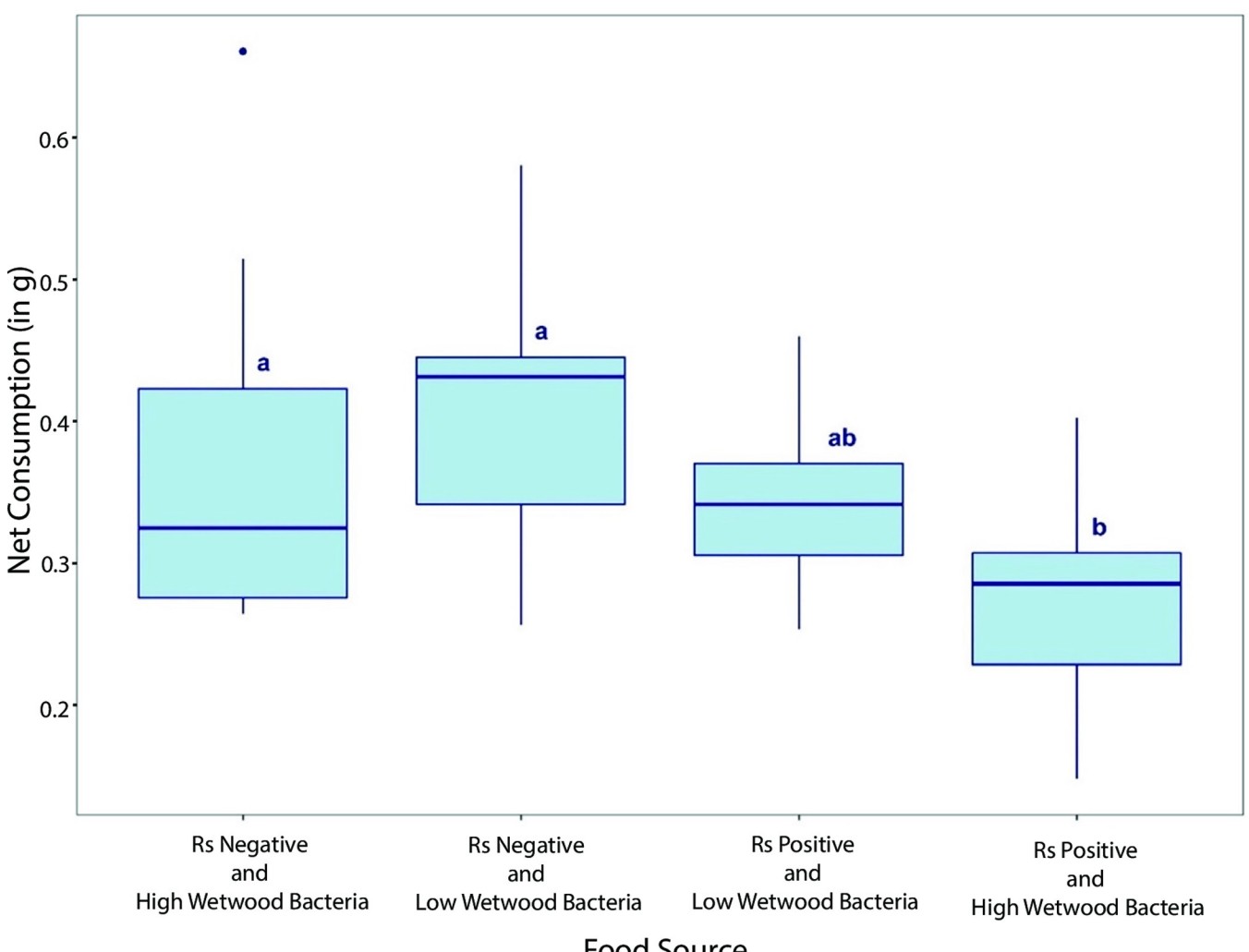

**Fig 4. Impact of *R. solanacearum* and wetwood bacterial loads of wood on consumtion by *N. takasagoensis* workers.** Net consumption (g) of wood pieces which tested positive or negative for species from the *R. solanacearum* complex (Rs) and contained low or high amounts of wetwood bacteria by *N. takasagoensis* workers. Different letters indicate significant differences determined by Tukey's Studentized Range Test for four-choice bioassay (S7 Table).

whether the termites would prefer wood with naturally low pathogen loads over wood with high amounts of wetwood bacteria and the presence of *Ralstonia*. In four-way choice tests between food sources consisting of natural wood pieces with different amounts of *Ralstonia* and wetwood bacteria, a significant effect of food source on net consumption by *N. takasagoensis* workers was observed (p = 0.01, R2 = 0.26, One-Way ANOVA, Fig 4). *Nasutitermes takasagoensis* workers consumed significantly less of the "*Ralstonia* positive and high amounts of wetwood bacteria" food source than the "*Ralstonia* negative and low amounts of wetwood bacteria" food source (p = 0.01), and the "*Ralstonia* negative and high amounts of wetwood bacteria" food source (p = 0.02, Tukey's Studentized Range Test, S7 Table). However, there was no significant difference in net consumption of the "*Ralstonia* negative and low amounts of wetwood bacteria," "*Ralstonia* negative and high amounts of wetwood bacteria," and "*Ralstonia* positive and low wetwood" food sources by *N. takasagoensis* workers. Additionally, there was no significant difference in the net consumption of "*Ralstonia* positive and low amounts of wetwood bacteria" food source and "*Ralstonia* negative and high amounts of wetwood

bacteria" food source (Fig 4 and S7 Table). These results suggest that termites tend to have a preference for wood with low or no amounts of IWTD pathogenic bacteria over wood containing high amounts of IWTD pathogenic bacteria.

## Termite consumption of wood pieces inoculated with different concentrations of *Ralstonia* versus saline control

In two choice tests, *N. takasagoensis* workers were given the option to feed on symptomless wood that was either inoculated with a known concentration of *R. solanacearum* or a saline control (symptomless wood without *Ralstonia*). The results showed that the consumption of the control wood was marginally higher than the consumption of wood inoculated with a $10^{-4}$ dilution (p = 0.09, $R^2$ = 0.08, n = 3 colonies x 5 replicates, ANOVA) and a $10^{-6}$ dilution of *Ralstonia* (p = 0.08, $R^2$ = 0.09, n = 3 colonies x 5 replicates, ANOVA), as shown in S2 Fig. However, at the lowest dilution ($10^{-8}$), no significant difference in consumption was observed (p = 0.11, $R^2$ = 0.11, n = 3 colonies x 5 replicates, ANOVA).

## Shift in bacteria composition of termites in no-choice tests

Two hundred forty termite workers fed on different concentrations of *Ralstonia* for different time periods were sequenced and 617 ASVs represented by 10,427,594 sequence reads and a minimum sequence depth per sample of 4,268 were obtained after removal of unassigned ASVs below the 97% identity threshold to SILVA database references. The sequencing-depth, sample-size and coverage used in no-choice tests were sufficient to capture most of the bacterial diversity present within these samples, as shown by alpha rarefaction curves (S3 Fig).

*Ralstonia* was not detected in any of the *N. takasagoensis* workers while they were feeding on *Ralstonia*-inoculated filter paper, regardless of the *Ralstonia* concentrations or feeding duration. However, *Ralstonia* was detected in six of the 240 worker samples (not in control samples) after a period of six days of *Ralstonia* feeding (Phase 1) and two days of feeding on filter paper without *Ralstonia* (Phase 2). Number of reads of *Ralstonia* ranged from 5,820 to 32,535 in those six samples, which represented less than 0.01% of the total bacteria community and only 2.5% of all samples.

While *Ralstonia* was not detected in the majority of the samples despite the forced feeding, a difference in the ranking of relative abundance of some phyla was observed in all samples between day 6 of Phase 1 and Phase 2 of the experiment (Fig 5). The average proportion of Fibrobacteres decreased by half, from 12% to 6%, which resulted in Fibrobacteres dropping from being the second most dominant phylum in Phase 1 to the fourth most dominant in Phase 2. The relative ranking of all other phyla remained the same, but their relative abundances changed. The proportion of Spirochaetes decreased, but to a lesser extent than Fibrobacteres, from 66% to 57%, while the proportion of Bacteroidetes increased from 8% to 12%. Firmicutes also increased from 6% to 9%, Proteobacteria increased from 4% to 9%, and Planctomycetes showed a slight increase from 2% to 3%. A significant difference between alpha diversity (Pielou's evenness, Faith's PD, ASV richness, Shannon diversity) (all p<0.009, Kruskal-Wallis ANOVA) and beta diversity (p = 0.001, PERMANOVA) of microbiota was observed between day 6 of Phase 1 and Phase 2 (S8 Table). There were 107 ASVs from four phyla—Fibrobacteres, Spirochetes, Bacteroidetes, and Firmicutes, that showed significant differences in abundance (q<0.05) between Phase 1 and Phase 2 (S9 Table). Fibrobacteres was the only phylum with all 14 ASVs enriched in Phase 1. Most (31 out of 49) ASVs from the phylum Spirochetes, mostly *Treponema* sp., were also found to be enriched in Phase 1. In contrast, most differentially abundant ASVs from the phyla Bacteroides (6 out of 8), Firmicutes (16 out of 21), Planctomycetes (6 out of 7), Acidobacteria (7 out of 7), and Synergistetes (1 out of 1)

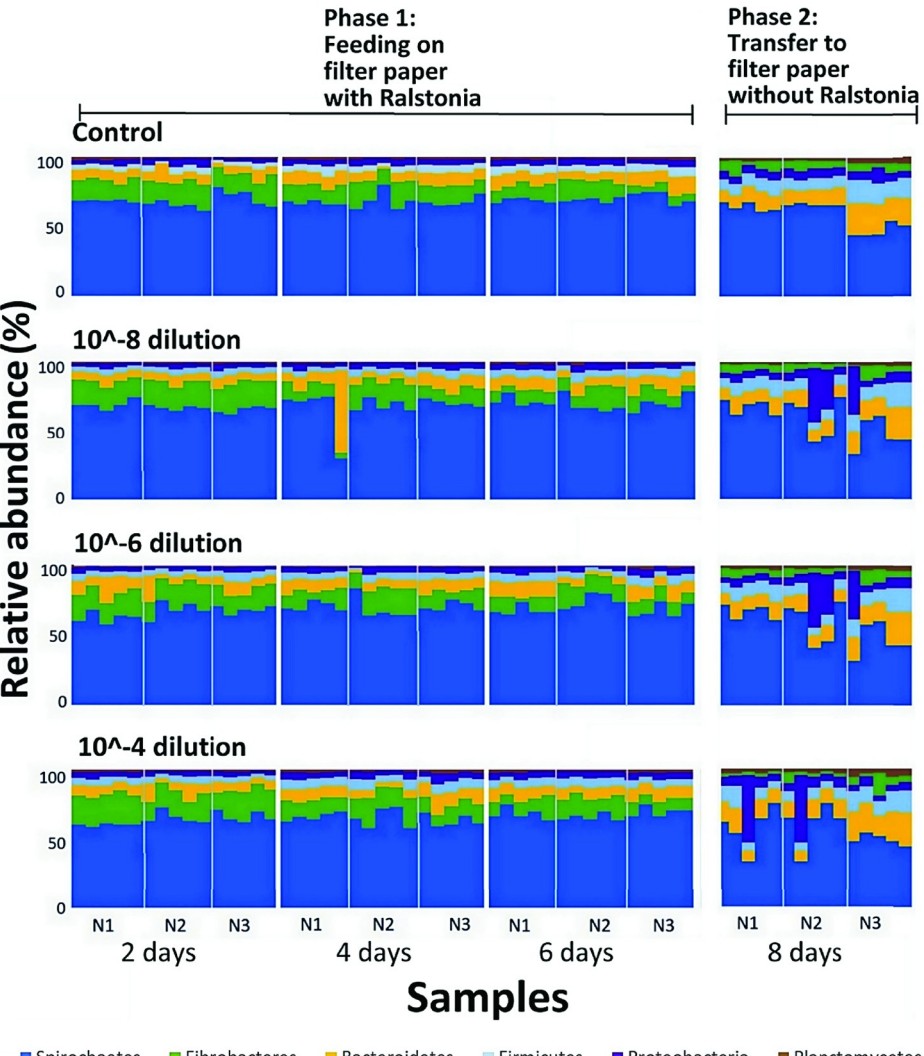

**Fig 5. Dynamics of bacterial phyla abundance in *N. takasagoensis* workers: Impact of *R. solanacearum* concentrations and feeding durations.** Taxa barplots showing the shift in relative abundance of bacterial phyla of termites between the two phases of the experiment with different *Ralstonia* concentrations (No *Ralstonia* control, $10^{-8}$, $10^{-6}$, and $10^{-4}$) and durations of feeding on *Ralstonia* inoculated filter paper (Phase 1: Termites were fed for 2, 4, and 6 days with different concentrations of *Ralstonia*. Phase 2: Termites fed for 6 days with *Ralstonia* were fed for two additional days on filter paper only for a total duration of the experiment of 8 days). There were five replicates for each *Ralstonia* concentration, time point and colony (N1, N2, N3).

were enriched in Phase 2. Next, we investigated if the microbiota shift was influenced by *Ralstonia* concentrations and feeding duration.

## Feeding on different *Ralstonia* concentrations did not impact the termite microbiome

Feeding termites with different *Ralstonia* concentrations ($10^{-4}$, $10^{-6}$, $10^{-8}$, and no *Ralstonia*) during Phase 1 had no significant effect on their bacterial communities in both Phase 1 (while feeding on *Ralstonia*) and Phase 2 (after transfer to *Ralstonia*-free filter paper) (S10 Table). The microbiome of termites in Phase 1 remained consistent regardless of *Ralstonia* concentrations, as there was no significant difference in alpha diversity measures (Pielou's evenness,

Faith's PD, ASV richness, Shannon diversity, all p≥0.41, Kruskal-Wallis ANOVA) or beta diversity (p = 0.27, PERMANOVA). Similarly, the microbiome of termites in Phase 2, that were initially fed on different *Ralstonia* concentrations during Phase 1 did not show significant differences (all p≥0.16, Kruskal-Wallis ANOVA for alpha diversity and p = 0.83, PERMA-NOVA for beta diversity) (Fig 5 and S10 Table). In addition, the change in microbiome composition observed between Phase 1 and Phase 2 was also observed in the control group without *Ralstonia*, suggesting that the presence or amount of *Ralstonia* did not contribute to the observed shift in the microbiome during the experiment (Fig 5).

### Impact of duration of feeding on termite microbiome

Considering that the presence of *Ralstonia* had no effect on the microbiome of termites, an analysis was conducted to determine whether the bacterial communities of termites were affected by the duration of the experiment during which they were fed filter paper in a laboratory setting. The results revealed that alpha diversity measures were affected in both phases of the no-choice test by the duration of feeding on filter paper, regardless of *Ralstonia* concentrations (Fig 6 and S11 Table). During Phase 1 (from 2 days to 6 days) of filter paper feeding with *Ralstonia*, there was a significant increase in the alpha diversity of the bacterial community, as measured by ASV richness, Faith's PD, Pielou's evenness, and Shannon diversity (all p≤0.04, Kruskal-Wallis ANOVA) (Fig 6 and S11 Table). Moreover, there was a significant increase observed in all alpha diversity measures from the beginning of Phase 1 to Phase 2 (from day 2 to day 6 of Phase 1 all p≤0.04, from day 6 of phase 1 to Phase 2, all p≤1.65E-04, Kruskal-Wallis ANOVA) (Fig 6 and S11 Table). Pairwise comparison of bacterial communities of termites (beta diversity) at 2 days, 4 days, and 6 days of Phase 1, and Phase 2 (8 days), detected significant differences (all p≤0.01, PERMANOVA) (S11 Table). The number of ASVs with differential abundance (q<0.05, ANCOM-BC) from six phyla (Fibrobacteres, Spirochetes, Bacteroidetes, Firmicutes, Proteobacteria, and Patescibacteria) nearly doubled between samples collected at 2 days and 4 days (18 ASVs) and those collected at 2 days and 6 days (31 ASVs) (S12 Table). A total of 8 ASVs from Phylum Fibrobacteres were enriched at 2 days as compared to 4 days while 11 ASVs were enriched at 2 days compared to 6 days, indicating a gradual decrease in Fibrobacteres within Phase 1 (S12 Table).

## Discussion

The aim of our study was to investigate if the termite *N. takasagoensis* is a pathogen vector responsible for the decline of ironwood trees in Guam. It was hypothesized that termites feeding on infected ironwood trees would carry pathogens and could transfer them to healthy trees while foraging [16, 17]. We utilized 16S rRNA gene amplicon sequencing data to examine the bacterial population found in *N. takasagoensis* worker samples collected from both healthy and diseased ironwood trees in Guam. The results suggest that *N. takasagoensis* workers are not vectors for pathogens linked to IWTD, as we did not find significant amounts of pathogens associated with IWTD in the termite samples, including *R. solanacearum*, *Klebsiella* species, and other potential pathogens like *Kosakonia*, *Enterobacter*, *Pantoea*, *Erwinia*, and *Citrobacter* [9, 12].

The lack of *R. solanacearum* in the bacterial microbiota of termite samples collected from ironwood trees in Guam was not likely due to any technical issues, such as sequencing failure, insufficient sampling depth, or a lack of reference sequences. We used primers that were tested for their ability to detect *R. solanacearum* by performing BLAST analysis against the NCBI GenBank database. The database, SILVA 132, used for the taxonomic assignment of ASVs obtained after sequencing the bacterial communities in our study, contained *R. solanacearum*

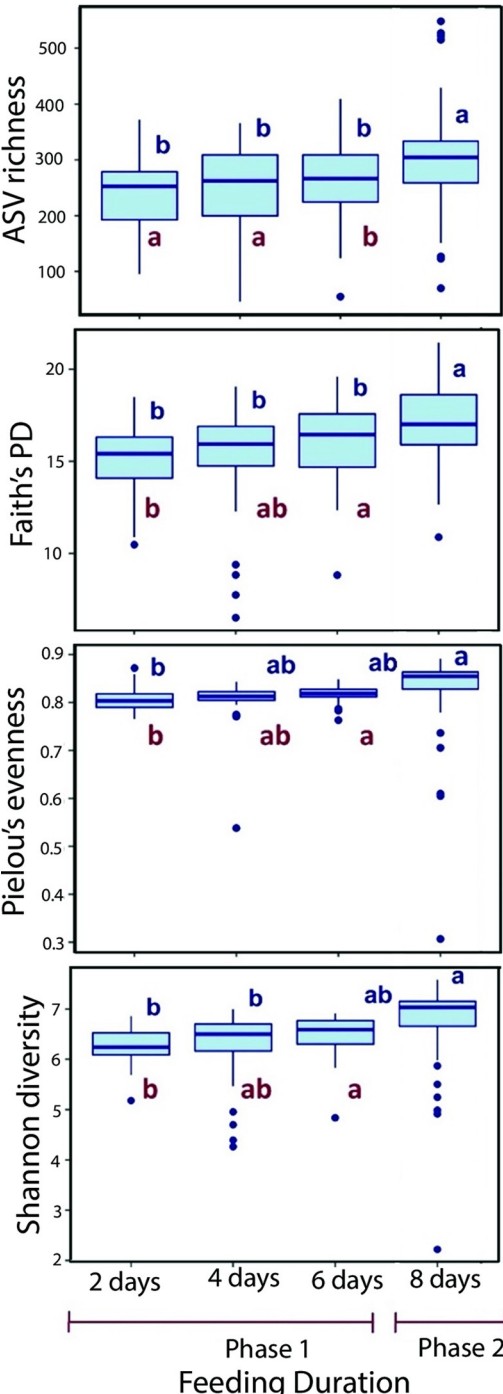

**Fig 6. Temporal variation in alpha diversity metrics of bacterial communities in worker guts.** Box plot shows differences in alpha diversity (ASV richness, Faith's PD, Pielou's evenness, Shannon diversity) of bacterial communities in termites during Phase 1 (represented by red letters below the boxes) and for the overall duration of feeding including Phase 1 and Phase 2 (represented by blue letters above the boxes).

and related strains of the *Ralstonia* complex. Additionally, the pure *Ralstonia* cultures employed in feeding experiments were verified to be of the same strain originally described

from IWTD-infested trees by means of sequencing, thus validating the strain and the identification methods. Moreover, during the examination of bacterial populations in these termite samples, various other genera that belong to the same family as the genus *Ralstonia* were detected. Despite not being found in freshly collected *N. takasagoensis* worker termites, *Ralstonia* was detected in workers that fed on *Ralstonia* inoculated filter paper in a no-choice feeding experiment. Furthermore, the rarefaction analyses indicated that the sequencing depth and sample effort were adequate in detecting the majority of bacteria linked to the termite samples.

Although *R. solanacearum* was not present, the taxonomic profiling was able to detect all of the core phyla observed in previous studies of *N. takasagoensis* microbiomes, including Spirochaetes, Fibrobacteres, Bacteroidetes, Firmicutes, Proteobacteria, Actinobacteria, and Margulisbacteria (TG3 phylum) [25–30]. Our study revealed Spirochaetes (mostly *Treponema* species) and Fibrobacteres as the most dominant phyla. The dominance of Spirochaetes and Fibrobacteres in the gut microbiome of termites was expected, as they contain obligate termite-specific symbionts, with Spirochaetes aiding in reductive acetogenesis, fermentation, and nitrogen fixation, and Fibrobacteres degrading cellulose and hemicellulose present in wood and plant material [23]. Spirochaetes have consistently been identified as the dominant phylum in the gut microbiome of higher termites, including *Nasutitermes* species [29, 30], and, specifically, in *N. takasagoensis* [25, 27, 28], regardless of the sequencing method being used. However, the abundance of Fibrobacteres has been reported to vary among previous studies on *N. takasagoensis* [25–28]. Hongoh et al. (2006) and Miyata et al. (2007) utilized clone-based methods and ranked Fibrobacteres as the second and fourth most abundant phylum, respectively, while Köhler et al. (2012) detected only a minor abundance (0.04%) of Fibrobacteres in the gut of *N. takasagoensis* workers using Pyrotag sequencing. In contrast to our study, none of the previous studies reported Fibrobacteres to be as abundant (41.38%) as Spirochaetes (48.16%) in the termite gut. Variations in primers, sequencing methods, and analytical techniques employed in the previous studies may have contributed to the observed variation in Fibrobacteres abundance. Moreover, the observed change in the abundance of Fibrobacteres in the no-choice feeding assay suggests that the abundance of Fibrobacteres can be influenced by various factors, including diet type, environmental conditions, and consequently, geographical location.

Based on our study, it is highly unlikely that *R. solanacearum* is present in the bodies of *N. takasagoensis* under natural conditions, as no sequences from the *R. solanacearum* species complex were detected among the 9.9 million sequencing reads obtained from 42 *N. takasagoensis* worker samples collected from ironwood trees in Guam, indicating that the probability of *Ralstonia* being present in these samples was less than 1 in 9.9 million bacteria. Possible explanations for the lack of *Ralstonia* and other IWTD pathogens in the *N. takasagoensis* worker samples are that termite workers do not forage or avoid feeding on infected wood altogether, or if they do consume bacteria from infected trees, the bacteria are unable to survive within the termite's body. Termites forage on or under the bark and on roots and are, therefore, less likely to encounter large amounts of xylem-residing *Ralstonia* and *Klebsiella* populations concentrated towards the center of the tree trunk [2, 10, 53, 54]. In addition, termites tend to show feeding preference for one type of wood over the other [55–57]. The feeding experiments in this study showed that termites prefer consuming healthy wood over wood containing IWTD pathogens, which would decrease the likelihood of pathogen contact even further. Even if termites encounter *Ralstonia* in the soil during foraging and nesting on infected trees, it is unlikely that the pathogen establishes a stable population in the termite body as our feeding experiments revealed that when *N. takasagoensis* workers were forced to consume high concentrations of *Ralstonia* on filter paper, *Ralstonia* was not detected for the first 6 days among the termite microbiome. It has been shown that termites usually consume

bacteria within 24 hours and rapidly transfer them to other members of the colony through trophallaxis and grooming [58, 59]; however, we did not find *Ralstonia* in the termite's body despite being fed on *Ralstonia* for more than 24 hours. This provides strong evidence that a healthy microbiota in termite body is refractory to invasion by *Ralstonia*. The absence of *Ralstonia* in the body of termites despite feeding on it could be due to unfavorable conditions in the gut preventing *Ralstonia* from growing as *N. takasagoensis* workers have an alkaline gut pH (6–10) and the growth optimum of species from the *R. solanacearum* complex is at a low pH (pH 4.5–5.5) [23, 60]. In addition, it has been reported that termites have innate, adaptive as well as social immunity that protects them against invasion by foreign microbes [61–65]. The indigenous gut symbionts of termites prevent foreign microbes present at feeding sites and food sources from colonizing the termite guts [66–70]. Furthermore, termite nests and tunnels have been found to contain specialized microbial communities producing antimicrobial peptides, which offer an additional layer of defense for the termite colony [64, 70–74].

It is interesting to note that certain *N. takasagoensis* worker samples did display considerable amounts of *Ralstonia* after consuming filter paper inoculated with *Ralstonia* for 6 days, followed by an additional 2 days of feeding on regular filter paper without *Ralstonia*. This increase in *Ralstonia* in some samples was accompanied by a shift in the microbiota in all samples, characterized by a decrease in Fibrobacteres, an increase in bacterial diversity, and a decline in termite health, which is most likely attributed to dysbiosis—an imbalance in the microbiota [75]. It is possible that dysbiosis weakened the protection provided by native gut symbionts, which in turn facilitated the invasion of foreign bacteria [76–78]. Consequently, *Ralstonia* was able to survive in some laboratory termite colonies, despite not being detected in the field samples.

The observed dysbiosis was not caused by *Ralstonia* ingestion, as experiments showed that *Ralstonia* concentrations had no effect on microbiota and termites that were not fed with *Ralstonia* also exhibited the same changes over time. Since in contrast to *Ralstonia* concentration, the duration of the experiment did show significant effects on the microbiota, we concluded that the dysbiosis might have occurred due to the removal of the termite workers from their natural environment and rearing them in the lab on an artificial diet, i.e., processed filter paper. This shift in microbiota observed after 8 days of feeding on filter paper was in accordance with results from previous studies which suggested that a shift in the microbiome might take place between one and two weeks when termites are fed on an artificial diet [27, 79–81]. Similar to our study, Miyata et al. (2007) observed a significant increase in the diversity and richness of microbiota in *N. takasagoensis* workers linked to a decrease in the relative abundance of Spirochaetes and Fibrobacteres after feeding workers for three weeks on artificial diets as compared to those feeding on wood. The majority of bacterial species in these two phyla are obligate symbionts in the guts of wood-feeding higher termites adapted to hydrolyzing complex lignocellulose from wood [26, 82, 83].

Although *Ralstonia* was not detected in samples of termites collected from both infested with *Ralstonia* and healthy ironwood trees, there were differences in the bacterial communities of termites collected from infested trees and those collected from *Ralstonia* negative trees. These differences in beta diversity were at least partly attributed to the differential abundances of a *Treponema* sp. from the phylum Spirochaetes and an unknown bacterium from the phylum Bacteroidetes, both of which were more abundant in termites attacking *Ralstonia* negative trees. These bacterial species are largely uncharacterized, but based on their described relatives, most bacterial species within these phyla are obligate symbionts of termites. *Treponema* species in termite guts are known to contribute to reductive acetogenesis and nitrogen fixation, while some bacteria within phylum Bacteroidetes play a significant role in the fermentation of sugars and uric acid degradation [77, 84, 85]. However, it is unclear how *Ralstonia* infection of trees

might affect the composition of termite microbiota. The consumption of *Ralstonia* is unlikely to change the termite microbiota because feeding experiments have indicated that *Ralstonia* alone is not responsible for shifts in microbial composition, and it is difficult for *Ralstonia* to thrive in a healthy termite. Nevertheless, there could be other factors confounded by or in interaction with *Ralstonia* infestation that affect beta diversity. The factors explored in our study only accounted for a small percentage (0.14%) of the variation in microbiota diversity, indicating that additional abiotic or biotic factors may have an impact on microbial composition and require further investigation in future studies.

Several aspects of the microbiome diversity of termites feeding on ironwood trees decreased with increased tree stress, such as a higher level of site management, an increased presence of declining, sick, and dead trees, and increased termite infestation in plots. However, phylogenetic diversity, a measure of the evolutionary relationships among microbial species, was higher in termites feeding on sick trees than those feeding on trees showing no symptoms of IWTD. The overall bacterial richness within the termite samples was not affected by tree stress or disease, suggesting that the number of species in the gut ecosystem is in a stable equilibrium that is limited by the number of ecological and functional niches available [23, 25, 86].

The wood of living trees contains antimicrobial defense mechanisms that protect the trees against the invasion of pathogens [87]. However, when tree health is impacted due to disease, the opportunity for pathogenic bacteria to colonize the tree tissues is increased leading to a change in the tree microbiome composition [88]. Thus, it is possible that stress, sickness, or death of the tree might change wood chemistry and, subsequently, the microbiome of the tree. A change in tree microbiome might have had direct or indirect impacts on the wood-feeding termite microbiota diversity in our study.

Based on our findings, we can exclude workers of *N. takasagoensis* as potential vectors of IWTD pathogens. However, it is possible that other termite species feeding on ironwood trees of Guam [21] might play some role in the transmission of these pathogens. For instance, members of the second most dominant termite species attacking ironwood trees, the Asian subterranean termite *Coptotermes gestroi*, have slightly acidic to neutral gut pH which can provide a favorable environment for *Ralstonia* to survive [89, 90].

Overall, termites are associated with IWTD [20]; however, it is still unknown whether termite infestation is one of the causes of IWTD or it is a consequence. The mechanical tree damage due to feeding of termites might be serving as a point of entry for the IWTD-associated pathogens. To understand the indirect association of termites with IWTD, the pattern of termite infestation at the different stages of IWTD severity needs to be studied. This would elucidate if association of termites are causal to IWTD, or if these termites simply serve as opportunistic feeders.

## Supporting information

**S1 Fig. Map of Guam (source: NASA earth observatory image created by Jesse Allen and Robert Simmon, using EO-1 ALI data provided courtesy of the NASA EO-1 team and the United States Geological Survey, taken on December 30, 2011, [URL: https://earthobservatory.nasa.gov/images/77189/guam]) showing ironwood tree sites from where 42 samples of *Nasutitermes takasagoensis* termites were collected.** Metadata for each termite sample can be found in S1 Table.
(TIF)

**S2 Fig. Net consumption (g) of wood pieces inoculated with $10^{-4}$, $10^{-6}$ and $10^{-8}$ *Ralstonia* dilution versus control by *N. takasagoensis* workers.**
(TIF)

**S3 Fig.** a) Sequence-based rarefaction curves of bacteria diversity showing the number of ASVs (ASV richness), Faith's phylogenetic distance and Shannon diversity indices for each of the 240 samples of Nasutitermes takasagoensis workers used in no-choice test plotted against sequencing depth. b) Sample-based rarefaction curves across all 240 samples with effective bacterial diversity for different metrics plotted against the number of samples. c) Coverage-based rarefaction curves across all 240 samples with effective diversity plotted against estimated sample coverage. Solid lines indicate intrapolation up to the actual sample size; dashed lines represent extrapolation to twice the sample size.
(TIF)

**S4 Fig.**
(TIF)

**S1 Table. Location-, tree-, and plot-related metadata for *Nasutitermes takasagoensis* termite samples.**
(XLSX)

**S2 Table. Number of reads across all samples, relative abundance of all the phyla associated with *N. takasagoensis* samples and the number of samples these phyla were observed in.**
(XLSX)

**S3 Table. List of all 462 ASVs with their SILVA 132 assignments detected in *N. takasagoensis* samples collected from ironwood trees in Guam.**
(XLSX)

**S4 Table. The most abundant 20 ASVs associated with *N. takasagoensis* samples according to total number of reads with their assignments in SILVA and NCBI GenBank, their total number of reads along with the number of samples the ASVs were observed in and the average number of reads and standard deviation per sample.**
(XLSX)

**S5 Table. Pairwise comparison of groups within the factors presence of *Ralstonia*, altitude and parent material using PERMANOVA (999 permutations) and PERMDISP (1,000 permutations).** Asterisks indicate significant effect.
(XLSX)

**S6 Table. ASVs with significant differential abundance in termites collected from *Ralstonia* positive versus negative trees, low versus high altitude, sand versus lime, and tuff versus lime parent material.** Significantly differential abundance was determined by ANCOM-BC at q value (adjusted p value) less than 0.05. W is the test statistic for determining differential abundance. Negative values for W depict that the ASV is significantly enriched in termites collected from trees that are a) *Ralstonia* negative compared to positive, or b) at high versus low altitude, or c) on lime compared to sand or tuff parent material and vice versa for positive W values. The enrichment of each ASV for a particular group is marked in blue.
(XLSX)

**S7 Table. Pairwise differences in consumption (above the diagonal) between four treatments with different natural wood pieces (along the diagonal) by *N. takasagoensis* workers and Tukey's Studentized Range Test results (below the diagonal) for the four-choice bioassay.**
(XLSX)

**S8 Table. Significant differences in alpha diversity (Pielou's evenness, Faith's PD, ASV richness, Shannon diversity) and beta diversity (Weighted Unifrac distance) between the bacteria communities of termites on day 6 of Phase 1 (feeding on filter paper with *Ralstonia*) and Phase 2 (2 days after transfer to filter paper without *Ralstonia*).** Asterisks indicate the significant differences.
(XLSX)

**S9 Table. Differentially abundant ASVs between the two phases of the experiment (Phase 1: After 6 days of feeding on filter paper with *Ralstonia* and Phase 2: After transfer to filter paper without *Ralstonia* for an additional 2 days).** Significance of differential abundance was determined by ANCOM-BC at q value less than 0.05. W is the test statistic for determining differential abundance. Positive values for W depict that the ASV is significantly enriched in Phase 1 (6 days) and negative values for W depict significant enrichment in Phase 2 (8 days). The phase in which an ASV is enriched is marked in blue for easy interpretation.
(XLSX)

**S10 Table. Lack of significant differences in alpha diversity (Pielou's evenness, Faith's PD, ASV richness, Shannon diversity) and beta diversity of the bacteria community of termites in both phases of the no-choice experiment with respect to the initial feeding with different *Ralstonia* concentrations (No *Ralstonia*, $10^{-8}$, $10^{-6}$, and $10^{-4}$).** Phase 1: Termites were fed for 2, 4, and 6 days with different concentrations of *Ralstonia*. Phase 2: Termites fed for 6 days with *Ralstonia* were fed for two additional days on filter paper only.
(XLSX)

**S11 Table. Alpha diversity (Pielou's evenness, Faith's PD, ASV richness, Shannon diversity) and beta diversity of bacteria community of termites with respect to duration of feeding in both phases of the no-choice experiment.** Phase 1: Termites were fed for 2, 4, and 6 days with different concentrations of *Ralstonia*. Phase 2: Termites fed for 6 days with *Ralstonia* were fed for two additional days on filter paper only.
(XLSX)

**S12 Table. Differentially abundant ASVs in Phase 1 (between 2 days and 4 days, and between 2 days and 6 days) of the experiment.** Significance of differential abundance was determined by ANCOM-BC at q value less than 0.05. W is the test statistic for determining differential abundance. Positive values for W depict that the ASV is significantly enriched at 2 days compared to 4 days or 6 days and vice versa for negative W values. The significant enrichment of each ASV for a particular time point is marked in blue.
(XLSX)

## Acknowledgments

The authors acknowledge Jin Tao and Ilgoo Kang for their help with the molecular work, Julia Hudson, Elizabeth Hahn, Alex Chingyan, and Ethan So for collecting the termite samples and helping in conducting bioassays. Thanks to Dr. Kelley Thomas and his team from the University of New Hampshire Hubbard Center for Genomic Studies for DNA sequencing and for providing the server for data analysis.

## Author Contributions

**Conceptualization:** Claudia Husseneder.

**Funding acquisition:** Robert Schlub.

**Methodology:** Junyan Chen, Claudia Husseneder.

**Supervision:** Claudia Husseneder.

**Writing – original draft:** Garima Setia.

**Writing – review & editing:** Robert Schlub, Claudia Husseneder.

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
