## [Decision Letter · Decision Letter 0]

11 Sep 2023

PONE-D-23-20524Taxonomic profiling of Nasutitermes takasagoensis microbiota to investigate the role of termites as vectors of bacteria linked to ironwood tree decline in GuamPLOS ONE

Dear Dr. Husseneder,

Thank you for submitting your manuscript to PLOS ONE. After careful consideration, we feel that it has merit but does not fully meet PLOS ONE’s publication criteria as it currently stands. Therefore, we invite you to submit a revised version of the manuscript that addresses the points raised during the review process.

We look forward to receiving your revised manuscript.

Kind regards,

Niraj Agarwala, Ph.D.

Academic Editor

PLOS ONE

4. We note that Figure S1 in your submission contain [map/satellite] images which may be copyrighted. All PLOS content is published under the Creative Commons Attribution License (CC BY 4.0), which means that the manuscript, images, and Supporting Information files will be freely available online, and any third party is permitted to access, download, copy, distribute, and use these materials in any way, even commercially, with proper attribution. For these reasons, we cannot publish previously copyrighted maps or satellite images created using proprietary data, such as Google software (Google Maps, Street View, and Earth). For more information, see our copyright guidelines: http://journals.plos.org/plosone/s/licenses-and-copyright.

a. You may seek permission from the original copyright holder of Figure S1 to publish the content specifically under the CC BY 4.0 license. 

5. Please keep your tables as part of your main manuscript and remove the individual files. Please note that supplementary tables (should remain/ be uploaded) as separate "supporting information" files

Reviewers' comments:

Reviewer's Responses to Questions

**Comments to the Author**

1. Is the manuscript technically sound, and do the data support the conclusions?

Reviewer #1: Yes

Reviewer #2: Yes

2. Has the statistical analysis been performed appropriately and rigorously? 

Reviewer #1: Yes

Reviewer #2: Yes

3. Have the authors made all data underlying the findings in their manuscript fully available?

Reviewer #1: Yes

Reviewer #2: Yes

4. Is the manuscript presented in an intelligible fashion and written in standard English?

Reviewer #1: Yes

Reviewer #2: Yes

5. Review Comments to the Author

Reviewer #1: Dear Sir,

Greetings! Thanks for the invitation to review the paper “Taxonomic profiling of Nasutitermes takasagoensis microbiota to investigate the role of termites as vectors of bacteria linked to ironwood tree decline in Guam” The paper is well written and suitable for the publication with some modification mentioned as under.

## Name of the family The ironwood tree (Casuarina equisetifolia), an indigenous agroforestry species in Guam, has been threatened by ironwood tree decline (IWTD) since 2002.

## (2, 5, 6, 7) Avoid too much reference only latest and relevant

## Intro and Material methods are well written please Rewrite the section We concluded that N. takasagoensis workers are not vectors for Ralstonia spp. or wetwood bacterial endophytes (Klebsiella, Pantoea, Enterobacter, Citrobacter, and Erwinia) previously identified in trees symptomatic for IWTD. Factors such as

Tree Health, Site Management, Plot Average Decline Severity, Proportion of Dead Tree in the Plot, Proportion of Trees with Termite Damage in the Plot, Presence of Ralstonia, and Altitude had a significant influence on bacteria diversity

Results and discussion i don’t feel any changes as its out of my expertise

Rest things are well written

Thanks

Reviewer #2: This is original piece of research work which provides great insight on how Nasutitermes takasagoensis acts as an potential vectors for IWTD pathogens and also provides on their action on these pathogens. It was delightful to read this piece of work.

6. PLOS authors have the option to publish the peer review history of their article (what does this mean?). If published, this will include your full peer review and any attached files.

Reviewer #1: **Yes: **HANUMAN SINGH JATAV

Reviewer #2: No

---

## [Author Response · Author response to Decision Letter 0]

6 Oct 2023

Dear Editor,

Thank you for considering our manuscript entitled “Taxonomic profiling of Nasutitermes takasagoensis microbiota to investigate the role of termites as vectors of bacteria linked to ironwood tree decline in Guam” for publication in PLOS ONE. We appreciate the time and effort you and the reviewers took to review our manuscript, and we are pleased to have an opportunity to respond to your concerns. 

Editor's Comments:

Response: We have reviewed the PLOS ONE style requirements and updated our manuscript to adhere to the specified file naming conventions. The manuscript now complies with the provided style templates.

2. In your Methods section, please provide additional information regarding the permits you obtained for the work.

Response: No permits were required to conduct the research.

3. PLOS requires an ORCID iD for the corresponding author in Editorial Manager on papers submitted after December 6th, 2016.

Response: In compliance with PLOS ONE's requirement, we have included the link to the ORCID iD of the corresponding author within the manuscript.

4. Figure S1 contains an image of a map that may be copyrighted. Replace the figure with a suitable alternative that complies with the Creative Commons Attribution License (CC BY 4.0).

Response: We have replaced the previous map with a map sourced from the NASA Earth Observatory, available in the public domain, as suggested. The new map has been appropriately cited in the figure caption.

5. Please keep your tables as part of your main manuscript and remove the individual files.

Response: We have integrated the tables into the main manuscript and removed the individual files. Additionally, supplementary tables have been provided as separate spreadsheets in the 'Supplementary Tables FileCH' file.

6. Please review your reference list to ensure that it is complete and correct. Avoid excessive references and prioritize the latest and relevant ones.

Response: We have carefully reviewed the reference list to ensure its completeness and correctness and reduced the number of references. 

Reviewer's Comments:

1. Avoid too many references; prioritize the latest and relevant ones.

Response: We appreciate the reviewer's feedback on the number of references used in the manuscript. We have carefully reviewed and revised the manuscript to prioritize the latest and most pertinent references, aligning with the aim of presenting up-to-date and relevant information. Some references were removed.

2. The Introduction and Materials and Methods sections are well written; however, the sentence "We concluded that N. takasagoensis workers are not vectors for Ralstonia spp. or wetwood bacterial endophytes (Klebsiella, Pantoea, Enterobacter, Citrobacter, and Erwinia) previously identified in trees symptomatic for IWTD" should be rewritten.

Response: We acknowledge the reviewer's positive feedback on the Introduction and Materials and Methods sections. We have carefully reevaluated the mentioned sentence and rephrased it for clarity and precision, in line with the constructive suggestion provided.

We are hopeful that the improvements made to our manuscript align well with the expectations of PLOS ONE. Your valuable insights have significantly contributed to refining our work. We eagerly await your feedback and the possibility of seeing our research published in your esteemed journal.

Best regards,

Garima Setia

Graduate Research Assistant, 

Louisiana State University, 

Baton Rouge, LA 70803

---

## [Editor Report · Decision Letter 1]

6 Dec 2023

Taxonomic profiling of Nasutitermes takasagoensis microbiota to investigate the role of termites as vectors of bacteria linked to ironwood tree decline in Guam

PONE-D-23-20524R1

Dear Dr. Husseneder,

We’re pleased to inform you that your manuscript has been judged scientifically suitable for publication and will be formally accepted for publication once it meets all outstanding technical requirements.

Kind regards,

Niraj Agarwala, Ph.D.

Academic Editor

PLOS ONE

Additional Editor Comments (optional):

Dear authors,

Thanks for modifying the manuscript as per reviewers suggestions.
---

## [Editor Report · Acceptance letter]

13 Dec 2023

PONE-D-23-20524R1 

PLOS ONE

Dear Dr. Husseneder, 

I'm pleased to inform you that your manuscript has been deemed suitable for publication in PLOS ONE. Congratulations! Your manuscript is now being handed over to our production team.

Kind regards, 

on behalf of

Dr. Niraj Agarwala 

Academic Editor

PLOS ONE